# Surveillance of Adenovirus and Norovirus Contaminants in the Water and Shellfish of Major Oyster Breeding Farms and Fishing Ports in Taiwan

**DOI:** 10.3390/pathogens11030316

**Published:** 2022-03-03

**Authors:** Viji Nagarajan, Jung-Sheng Chen, Gwo-Jong Hsu, Hsin-Pao Chen, Hung-Chun Chao, Shih-Wei Huang, I-Sen Tsai, Bing-Mu Hsu

**Affiliations:** 1Department of Earth and Environmental Sciences, National Chung Cheng University, Chiayi County 621301, Taiwan; mathumitha08@gmail.com (V.N.); ekman60@gmail.com (H.-C.C.); ian61416@gmail.com (I.-S.T.); 2Department of Medical Research, E-Da Hospital, Kaohsiung 824410, Taiwan; ed113187@edah.org.tw; 3Division of Infectious Diseases, Ditmanson Medical Foundation, Chia-Yi Christian Hospital, Chiayi City 60002, Taiwan; cych01347@gmail.com; 4Division of Colon and Rectal Surgery, Department of Surgery, E-DA Hospital, Kaohsiung 824410, Taiwan; ed102430@edah.org.tw; 5School of Medicine, College of Medicine, I-Shou University, Kaohsiung 824410, Taiwan; 6Institute of Environmental Toxin and Emerging Contaminant, Cheng Shiu University, Kaohsiung 833301, Taiwan; envhero@mail.csu.edu.tw; 7Center for Environmental Toxin and Emerging Contaminant Research, Chen Shiu University, Kaohsiung 833301, Taiwan; 8Center for Innovative on Aging Society, National Chung Cheng University, Chiayi County 621301, Taiwan

**Keywords:** adenovirus, norovirus, oyster breeding farms, phylogenetic analysis, shellfish

## Abstract

The enteric viruses, including adenovirus (AdVs) and norovirus (NoVs), in shellfish is a significant food safety risk. This study investigated the prevalence, seasonal occurrence, genetic diversity, and quantification of AdVs and NoVs in the water and cultured shellfish samples at the four major coastal oyster breeding farms (COBF), five major fishing ports (FP), and their markets in Taiwan. The AdVs/NoVs in the water and shellfish samples were isolated by the membrane filtration and direct elution methods. The RNA of NoVs was reverse-transcribed into complementary DNA through reverse transcription reaction. Further NoVs and AdVs were detected using nested PCR. A higher detection rate was recorded in the low-temperature period than high-temperature. Detection difference was noted between nested PCR and qPCR outcomes for AdVs. The total detection rate of AdVs was higher in the water samples (COBF-40.6%, FP 20%) than the shellfish samples (COBF-11.7% and FP 6.3%). The AdVs load in the water and shellfish samples ranged from 1.23 × 10^3^ to 1.00 × 10^6^ copies/L and 3.57 × 10^3^ to 4.27 × 10^4^ copies/100g, respectively. The total detection of NoVs was highest in the water samples of the FP and their market shellfish samples (11.1% and 3.2%, respectively). Genotyping and phylogenetic analysis were identified as the prevalent AdVs and NoVs genotypes in the water and shellfish samples: A species HAdVs serotype 12; F species HAdVs serotype 41; and C species PAdVs serotype 5 (NoVs GI.2, GI.3 and GII.2). No significant differences were observed between the presence of AdVs, and all of the water quality parameters evaluated (heterotrophic plate count, water temperature, turbidity, pH, salinity, and dissolved oxygen). The virus contamination occurs mainly due to the direct discharge of domestic sewage, livestock farm, and fishing market wastewater into the coastal environment. Thus, this study suggested framing better estuarine management to prevent AdVs/NoVs transmission in water and cultured/distributed shellfish.

## 1. Introduction

Enteric viruses are frequently detected in environmental waters, including coastal and estuarine regions, and may be associated with water contamination by fecal pollution. Moreover, the discharge of effluents from the wastewater treatment plants directly into the coastal ecosystem may contaminate water sources and lead to the continuous release of human viruses into the coastal environment, which may cause serious human health hazards. Certainly, these viruses can survive for a long time either in the water or by attaching to particulate matter and subsequently getting accumulated in sediments [1]. Consequently, viruses that desorb from sediment may be transported through the water to non-contaminated areas [1]. Generally, shellfish are cultured in inshore coastal environments, which may pollute with significant quantities of fecal contaminants from urban runoff, point-source discharges, and disposal from boats [2]. They can filter the neighboring water from their surroundings by their bioaccumulation capacity for feeding. Moreover, they will concentrate and accumulate/retain enteric viruses derived from sewage contamination in their edible tissues and act as a vehicle for transmitting foodborne pathogens [2]. Despite the depuration process used to clean the harvested shellfish, it does not appear to be an effective process to remove the contaminant viral particles [3]. Consumption of contaminated shellfish can lead to many enteric virus outbreaks since shellfish is often consumed raw or half-cooked [4,5]. 

Enteric viruses include a diverse group of pathogens responsible for various diseases such as respiratory illness, gastroenteritis, and conjunctivitis in humans through ingestion of contaminated water or food [6]. Among them, adenovirus (AdVs) and norovirus (NoVs) are important pathogens concerning their transmission routes. Adenovirus is a non-enveloped double-stranded DNA virus that belongs to the family Adenoviridae. It is often linked with respiratory illness, gastroenteritis, and neurological diseases. Moreover, AdVs can persist in the environment for a longer time and are considered one of the potential human viral indicators present in fecal-contaminated waters [7]. Though AdVs infection is typically asymptomatic in healthy humans, its presence in the shellfish may prove to be a useful indicator of viral contamination [3]. NoVs is a non-enveloped single-stranded RNA virus that belongs to the family Caliciviridae with a diameter of approximately 38 nm [8,9]. These viruses are often detected in water bodies, including marine environments, rivers, and recreational water bodies [10,11,12,13,14]. Eventually, these viruses are bio-accumulated by the shellfishes from contaminated waters via their filter-feeding mechanism, and they are retained for prolonged periods than fecal indicator bacteria [15]. NoVs are known as the major cause of outbreaks of acute gastroenteritis and to be responsible for 60–80% of foodborne outbreaks worldwide [16]. It is one of the principal agents detected in oyster-associated outbreaks of gastrointestinal disease observed in many countries [17,18,19]. Monitoring the predominance of enteric viruses in the sea/port water and shellfish requires precise detection and quantification methods. Molecular techniques, such as polymerase chain reaction (PCR), offer sensitive and rapid detection of viruses in various environmental samples. Nested-PCR can detect even a small number of viruses in samples and provide a high level of sensitivity. 

Seafood, including fish and oysters, is an important food source worldwide. In Taiwan, generally, the shellfish are cultured in the breeding farms and distributed to the nearby fishing markets. As to mariculture, oyster farming significantly contributes to the livelihood of the seaside community [20]. However, contamination of coastal and estuarine waters by fecal pollution through the nearby urban runoff may deteriorate the commercial utilization of shellfish and eventually cause serious human health risks. The presence of fecal indicator bacteria in shellfish is usually used to assess the quality [21]. Fecal contamination of shellfish growing areas is likely to increase in the future due to rising human populations. Thus, the development of sewer system coverage is necessary for the future under the growing population. Besides, it is essential to meet the market requirements to supply safe shellfish products to consumers [22]. 

In this context, a surveillance study was conducted to examine the prevalence, distribution, seasonal occurrence, genetic diversity, and quantification of AdVs and NoVs in the cultured shellfish samples of the coastal oyster breeding farms. Moreover, to track the effect of shellfish distributed to various markets, shellfish samples were also collected from the nearby fishing ports markets. To spot the viral contamination sources, water samples of the oyster breeding farms and fishing ports were collected. This water is used for all the fishing activities, such as breeding and depuration. Additionally, the correlation between the physical and microbial water quality parameters and the presence of AdVs was investigated. 

## 2. Results

### 2.1. Detection and Quantification of Adenovirus and Norovirus in the Water and Shellfish Samples

The total detection rate of AdVs and NoVs in the water and shellfish samples from the fishing ports (FP) and COBF (coastal oyster breeding farms) are summarized in Table 1 and Table 2. The total detection rate of AdVs was the highest in the water samples of COBF (40.6%) and FP (20%). It was followed by the shellfish samples of COBF (11.7%) and FP (6.3%). The water samples of FP didn’t exhibit any seasonal variation regarding AdVs detection (20% detection at each period). But the COBF water samples recorded the highest rate of detection in the low-temperature period (56.5%), followed by the high-temperature (47.8%) and warm-temperature period (17.4%). Regarding shellfish, the FP market samples didn’t show any difference in the warm and high-temperature periods (9.5%). But AdVs was not detected in the shellfish samples during the low-temperature period. In the case of COBF, the cultured shellfish samples recorded the highest AdVs detection in the low-temperature (20%) followed by the warm-temperature period (15%). However, AdVs was not detected in the shellfish samples from the above site in the high-temperature period. Besides, the detection rate of AdVs in each FP and COBF are examined. Regarding water samples, among the FP, the Dongshi (DS) FP water samples have recorded the highest rate of total AdVs detection (55.5%), followed by Nanliao (NL) (33.3%) and Fuji (FJ) FP (11.1%). However, the Wuchi (WC) and Budai (BD) FP water samples were not AdVs detected throughout the study time. Among the COBF water samples, the highest AdVs detection was recorded in the Wanggong (WG) (66.6%) followed by Kouhu (KH) (58.3%), Dongshi-Budai (DSBD) (46.7%), and Chiku (CK) COBF (14.3%). Concerning the shellfish samples, none of the market shellfish samples contained detected AdVs except the WC (33.3%) and FJ FP (11.1%). Likewise, the cultured shellfish samples from all the COBF were not detected AdVs except for DSBD (16.7%) and CK (13.3%). 

Compared to AdVs, NoVs was detected at a lower rate in all the tested samples. The total detection of NoVs was highest in the water samples of the FP and their market shellfish samples (11.1% and 3.2%, respectively). However, the COBF cultured shellfish (1.7%) and water samples (0%) have exhibited a minimal and no detection of NoVs, respectively. Besides, the water samples of the FP have recorded the highest detection in the low-temperature (20%) followed by the warm-temperature period (13.3%). However, NoVs was not detected in the water samples during the high-temperature period. Likewise, the shellfish samples from the FP had no detected NoVs throughout the study time, except in the warm-temperature period (9.5%). Besides, the water and shellfish samples from the COBF didn’t detect NoVs contamination in the entire study except for its detection in the shellfish sample in the high-temperature period (5%). Regarding individual sampling sites, the NL FP (33.3%) water samples recorded the highest detection rate, followed by WC FP (22.2%). The remaining water samples from the FP (FJ, BD, and DS) did not present NoVs. Similarly, no NoVs was detected in the water samples from all the COBF during the entire study time. Concerning the shellfish samples, only the WC (11.1%) and BD FP (3.7%) recorded the least detection, while the remaining samples were not detected NoVs. Likewise, the cultured shellfish samples from all the COBF didn’t detect NoVs except CK (6.7%).

Detection difference was noted between nested PCR and qPCR outcomes for AdVs (Table 1 and Table 2). The variation in detection and quantification may be due to the adaption of the various PCR protocols and primers used. Regarding the water samples, none of the FP recorded a detectable AdVs concentration during the low-temperature period (Table 1). Likewise, during the warm-temperature period, except for the NL FP (5.25 × 10^4^ copies/L), none of the FP recorded quantifiable AdVs concentration. However, none of the FP water samples has detected measurable AdVs concentration during the high-temperature period. Besides, the highest AdVs concentration was recorded in the water samples of KH COBF during the low-temperature period (7.94 × 10^4^ copies/L), followed by WG (6.03 × 10^3^ copies/L) and DSBD (1.23 × 10^3^ copies/L). Similarly, in the warm-temperature period, the highest AdVs concentration was noted in the water samples of WG (1.63 × 10^4^ copies/L) and DSBD (1.53 × 10^4^ copies/L), while the remaining COBF didn’t record measurable AdVs concentration (KH and CK) during the warm-temperature period. However, none of the COBF water samples showed quantifiable AdVs concentration during the high-temperature period except the DSBD COBF (1.00 × 10^6^, 6.25 × 10^5^, 7.78 × 10^4^, 8.32 × 10^4^, and 1.35 × 10^4^ copies/L). Regarding the shellfish samples, none of the shellfish samples of the FP markets has detected measurable AdVs concentration in the low and high-temperature period (Table 2). In the warm-temperature period, the shellfish samples from only the FJ FP (4.27 × 10^4^ copies/100g) were quantified AdVs, while the remaining FP didn’t record a quantifiable AdVs load. Besides, the highest AdVs concentration was exhibited in the shellfish samples of the DSBD (7.03 × 10^3^ copies/100g) and CK COBF (3.57 × 10^3^ copies/100g) during the warm-temperature period, while the remaining COBF were not detected with measurable AdVs load (WG and KH). Likewise, none of the shellfish samples from the COBF quantified the AdVs concentration during both the low and high-temperature periods.

### 2.2. Genotyping and Phylogenetic Analysis of Adenovirus and Norovirus

In this study, the AdVs/NoVs positive samples were subjected to nucleic acid sequencing and species identification. AdVs, HadVs, and PAdVs were predominant in the water and shellfish samples found in FP and COBF (Table 3). The following three serotypes were the most prevalent: A species HAdVs serotype 12; F species HAdVs serotype 41; and C species PAdVs serotype 5. The frequency of various HAdVs and PAdVs serotypes was studied. Regarding the water samples, the highest number of HAdVs 41 was detected in the NL FP (3/3), followed by FJ (1/3) and DS FP (1/3). While, in the COBF water samples, the detection was as follows; DSBD (HAdVs 41–4/10; HAdVs 12–1/10); CK (HAdVs 41–2/7; HAdVs 12–1/7); and KH (HAdVs 41 -1/4). Concerning the shellfish samples, the highest amount of HAdVs 41 was detected in the WC (3/6) and FJ FP (1/3). In the COBF shellfish samples, the detection was as follows; DSBD (HAdVs 41–3/20; HAdVs 12–1/20); and CK (HAdVs 41–1/5). Apparently, none of the water samples of FP showed PAdVs 5 throughout the study time except DS FP (4/6). In the COBF water samples, the PAdVs 5 detection was DSBD (9/20), KH (6/12) and WG COBF (4/6). None of the FP and COBF shellfish samples showed PAdVs at the entire study time, except DSBD (1/10) and CK COBF (1/5). Regarding NoVs, the GI and GII genogroups were prevalent in the water and shellfish samples of FP and COBF (Figure 1). Three genotypes, NoVs GI.2, GI.3, and GII.2 were prevalent in the samples, which are often associated with foodborne outbreaks. However, the most prevalent genotypes, such as GII.4 and GII.17, were not detected in the water and shellfish samples. NoVs GII.2 exhibited the highest detection frequency in the water samples of NL (3/6) and WC FP (2/3). While the shellfish samples of WC FP (1/3) and CK COBF (1/5) showed NoVs GI.2 and BD showed GI.3 (1/3).

### 2.3. Water Quality Analysis 

Statistical analysis was performed to determine the correlation between the presence of AdVs and the water quality indicators. The results of nonparametric tests for AdVs (Mann-Whitney test) are shown in Table 4. No significant difference was observed between the AdVs-positive and negative samples with the water quality parameters such as heterotrophic plate count, temperature, turbidity, pH, salinity, and dissolved oxygen (Mann-Whitney U test, *p* > 0.05).

## 3. Discussion

This study identified the viral etiological agents in the major COBF and FP in Taiwan. It characterized the presence, distribution, genotypes, and quantification of AdVs and NoVs in the water and shellfish samples of the COBF and FP. In this present study, the water and shellfish samples of FP and COBF showed AdVs, and the total detection rate was as follows: COBF-W (40.6%), FP-W (20%), COBF-S (11.7%), and FP-S (6.3%). Previous studies conducted in Taiwan have shown that an abundance of AdVs was found in sewage-contaminated coastal water (34.3%) and river water (30.8%) [23,24]. Likewise, the samples were examined for NoVs occurrence, but the detection rate was lower than AdVs. Comparatively, the total detection of NoVs was found to be higher in the FP (W—11.1%, S—3.2%) than COBF (S—1.7%, W—0%), which is associated with the clinical illness of humans consuming contaminated food. Exposure to these enteric viruses may happen through direct ingestion of contaminated shellfish or by recreational exposure to contaminated water [3,6,25]. It is evident that 14–23% of NoVs infections are attributed to foodborne transmission worldwide [26] and particularly gastroenteritis outbreaks associated with the ingestion of NoVs contaminated shellfish in the USA, New Zealand, Canada, and Australia [2,25,27,28,29,30]. Indeed, the incidence of AdVs/NoVs found in the harvested shellfish samples suggests that these viruses may be endemic in sewage polluted coastal environments.

Concerning individual sampling sites, the water sample of DS FP recorded the highest rate of AdVs detection (55.5%); since the DS FP is located exactly near the estuarine area of the Puzi River. A previous study in the Puzi river evidenced the predominance of HAdVs in the river water (34.3%). The viral contamination is related to the discharge of Chiayi city municipal sewage and livestock wastewater into the river [23]. Thus, the flow-out of the Puzi River could be associated with the AdVs contamination of the nearby DS FH. Moreover, the AdVs detection in the NL and FJ FP may be associated with human fecal contamination. However, that the water samples of WC and BD FP did not exhibit AdVs contamination may be due to the absence of polluted water discharge near the vicinity of the FP. By contrast, the shellfish samples collected from the WC FP recorded notable AdVs. The concentration of human enteric viruses in marine sediments is nearly 10–10,000 times higher than that of overlaying marine water [31]. The adsorption of the enteric virus by the marine sediment may protect them from environmental inactivation and would result in unexpected resuspension [31]. Further, the enteric viruses desorbed from the polluted marine sediment may be transported through the water to unpolluted zones [1]. Besides, the market shellfish samples of FJ FP showing AdVs may be connected with the use of fecally-contaminated FP water for depuration and allied activities. The release of effluents from wastewater treatment plants into the marine environment can contaminate waters used for recreation purposes, as well as adjacent shellfish beds, which in turn affect their commercial utilization. Besides the effluents, biosolids from the wastewater treatment plants have been shown to contain infective enteric viruses and could cause potential human illness [32]. The direct discharge of wastewater from the nearby coastal markets and effluents from the fish processing plants into the coastal and nearshore environment may also be a possible source of contamination [33]. Likewise, the water sample from the KH COBF detected AdVs at a higher rate since it is situated along the coastline near the estuarine areas of the Beigang River. A prior study conducted in the Beigang river proves that AdVs contamination is associated with fecal contamination and thus to anthropogenic activities [24]. The river outflow may connect with the contamination and AdVs dominance in the KH COBF. Likewise, the water samples of DSBD COBF showed AdVs presence likely due to the sewage and livestock farm wastewater discharge in the Puzi River basin, which is situated near the oyster breeding farm [23]. Similarly, AdVs detection in the water samples of WG and CK COBF may be allied with urban runoff in the surroundings. Hence, the shellfish samples from the above-contaminated sites (DSBD and CK) were infected with AdVs, which may be due to the usage of fecally contaminated water for culturing purposes and other related activities. Being filter feeders, oysters can pump large volumes of water across their gills, which leads to a concentration of waterborne contaminants in their tissues. Moreover, the oysters accumulate enteric viruses from the culturing water and concentrate them in their gills and digestive glands. The accumulated enteric viruses can be measurable for 6–10 weeks after contamination, and it remains infective for 2–4 weeks in their tissues [34]. A prior study proved that residual enteric virus contamination in shellfish might persist even after a week of depuration [15]. Moreover, the depurated shellfish typically retain higher levels of enteric viruses than coliform bacteria [3]. 

Regarding NoVs, the NL and WC FP recorded the highest detection rate. A previous study reported that the Wu River was contaminated with fecal pollution and human activities. It is confluent with the seawater where the WC FP is located [24]. The shellfish samples collected from the fecally contaminated WC FP accumulated the virus in their tissues via a feeding mechanism. However, in this study, the water and shellfish samples from all the COBF free from NoVs contamination or the concentration may be less than the detection limit during the entire study time except CK. The shellfish collected from the CK COBF recorded the least number of NoVs detection, which may be associated with the sedimented viral loads [31]. NoVs can survive for an extended period in water, particularly in sediment/particles, and the virus is taken up by oysters as part of their feeding process [35]. Generally, the oysters accumulate NoVs to a concentration higher than that in the growing water, and it can remain in the sample for even two months after the contamination [36]. Particularly, the presence of histo-blood group types of antigens in shellfish tissues, which is a NoVs receptor, may result in the ineffective removal of NoVs and lead to its enhanced persistence [37]. The virus does not multiply in food but remains viable for extended periods. Besides, the NoVs accumulation potential within the oysters may depend on NoVs genotypes and the oyster species grown [38].

The prevalence of AdVs/ NoVs in water and shellfish samples were highest in the low-temperature period. An increased prevalence of AdVs in the winter season (low-temperature) has been reported by a prior study conducted in Greece and Sweden (73% and 60%, respectively) [39]. As the survival rate of AdVs is inversely proportional with temperature, the occurrence is higher in the low-temperature than in the high-temperature period [40]. Similarly, the incidence of NoVs in water and shellfish samples was the highest in the low-temperature period. Prior studies conducted in Japan, South Korea, and Spain have reported that the incidence of NoVs was the highest in the winter season than in spring [41,42,43]. Moreover, most of the NoVs outbreaks are reported in the winter season worldwide due to their better survival/spread in low-temperature [44,45]. The nested-PCR and qPCR were used to detect and quantify AdVs/NoVs in the samples. The nested PCR improves the assay’s sensitivity, and it can detect very low numbers of viral particles in the sample. Nevertheless, nested PCR outcomes of most water and shellfish samples were positive. Only a few samples detected notable AdVs concentration in qPCR assay. The qPCR assay exhibited the highest AdVs concentration in the water samples of NL FP and WG, KH, DSBD COBF. A prior study conducted in the Puzi River recorded a notable AdVs concentration in the winter season (2.8 × 10^3^ copies/L), which may be associated with the higher AdVs detection in the DSBD COBF [23]. Moreover, the water samples of Beigang River exhibited a higher AdVs concentration in qPCR assay, which possibly links with the notable AdVs concentration in the KH COBF. 

Genotyping studies revealed that HAdVs and PAdVs was the predominant genotype found in the water samples. The prevalence of HAdVs in the samples was generally linked with the fecal contamination and risk of contamination by NoVs [46]. Particularly, the F species AdVs serotype 41 causes gastroenteritis in humans. It may infect humans through ingestion of contaminated water during recreational activities or by consuming shellfish harvested from contaminated waters [6]. These AdVs genotypes may be shed in feces several months or even years after infection [47]. The higher HAdVs 41 detection was noted in the DS FP, DSBD COBF, and WC FPM samples. It may be associated with HAdVs 41 detection in the Puzi and Wu River, whose estuarine is situated near the DS, DSBD, and WC FP, respectively [23,24]. Likewise, the KH COBF detected HAdVs 41, which may be linked with the abundance of this particular serotype in the Beigang River located nearby [24]. Besides, viral contamination may occur when human excrement containing the virus flows into nearby FP and COBF through urban runoff. Thus, humans in contact with the contaminated port/farm water and consuming the contaminated shellfish are at an increased risk of infection with HAdV 41. It is one of the major etiological agents for most cases of gastroenteritis in children [48]. The infected people may act as carriers by shedding viruses in their stools without showing any visible symptoms [49]. Besides, the DS FP, WG, KH, and DSBD COBF have detected PAdVs 5 in their water samples, which may be linked to the outflow from the nearby rivers contaminated with the livestock farm wastewater [23,24]. PAdVs 5 cause gastrointestinal and multifactorial respiratory disease in swine [50], and this serotype is a potential indicator of swine fecal contamination [51]. Regarding NoVs, the GI and GII genogroups were prevalent in the water and shellfish samples. The environmental stability of GI and GII genogroups in water possibly facilitates their transport from the contamination source to the receiving water. NoVs GI and GII, the genotypes that cause gastroenteritis in humans, have been detected in polluted surface waters and shellfish worldwide [52,53,54]. NoVs GI genogroups are more responsible for waterborne outbreaks than GII genogroups [55]. This genotype has higher stability in water and a differential accumulation efficiency in binding to histo-blood-group-antigen in oysters [37]. NoVs GI genogroup binds to strain-specific carbohydrate moieties present in the oysters, undifferentiated from HBGA A antigen [56]. The detected NoVs genotypes GI.2, GI.3, and GII.2 are mostly related to clinical infections.

In this study, no significant differences were observed with the presence of AdVs concerning heterotrophic plate count, water temperature, turbidity, pH, salinity, and dissolved oxygen. This finding was in line with the previous findings, where no significant differences were observed between the HAdVs positive and negative samples with water quality indicators [24]. The variables recognized as influencing virus viability include temperature, pH, salinity, presence of solids, and indigenous microbiota [57]. Though in previous studies a significant association was observed for the presence of AdVs with various physicochemical parameters, the association with microbiological parameters was not reported [35].

## 4. Materials and Methods

### 4.1. Sampling Locations, Sample Collection, and Water Quality Analysis

Approximately 1 L of water and *Crassostrea angulate* shellfish (20–25 oysters) samples were collected at each of five major fishing ports (FP) and their respective markets around Taiwan; namely Fuji (FJ), Nanliao (NL), Wuchi (WC), Dongshi (DS,) and Budai (BD) fishing ports. Similarly, water and shellfish samples were collected at four coastal oyster breeding farms (COBF) around Taiwan; namely Wanggong (WG), Kouhu (KH), Dongshi Budai (DSBD), and Chiku (CK) between December 2016 and August 2017 in the following three-periods: low-temperature (December-February: 18–21 °C), warm-temperature (March-May:18–26 °C), and high-temperature (June–August:28–32 °C). An overview of all the sampling locations is shown in Figure 2. The samples were collected at different points within each sampling site, and the number of samples collected from the FP and COBF were varied based on the sample availability. The water sampling details of FP and COBF are as follows; FJ-3/15, NL-3/15, WC-3/15, DS-3/15, BD-3/15 and WG-2/23, KH-4/23, DSBD-10/23, and CK-7/23 per each period, respectively. The shellfish sampling details of FP and COBF are as follows; FJ-3/21, NL-3/21, WC-3/21, DS-3/21, BD-9/21 and WG-2/20, KH -3/20, DSBD-10/20, and CK-5/20 per each period, respectively. A total number of 45 (15/period) and 69 (23/period) water samples, and 63 (21/period) and 60 (20/period) shellfish samples, were collected from the FP and COBF, respectively. The collected samples were transferred and stored at controlled temperature until further molecular analysis. 

The water samples were subjected to various quality assessments. The physical water quality parameters, including water temperature, salinity, pH, and dissolved oxygen, were measured in real-time using a portable multi-parameter meter (HI9828, Hanna Instruments, Woonsocket, RI, USA). Turbidity was examined using a turbidimeter (HACH Co., Loveland, CO, USA). Microbial water quality analysis was done according to the standard protocol for examining water [58]. The heterotrophic bacteria counts were measured using the spread plate method (Methods 9215C). 

### 4.2. Virus Concentration, Pre-Treatment of Shellfish, Nucleic Acid Extraction, and Reverse Transcription

The collected water sample (1 L) was vacuum-filtered through a 47 mm GN-6 membrane with a pore size of 0.45 µm (GN- Metricel PALL, New York, NY, USA) [59]. Next, the membranes were scraped, and the collected material was washed with 20 mL of phosphate-buffered saline (PBS; 7.5 mM Na_2_HPO_4_, 3.3 mM NaH_2_PO_4_, 108 mM NaCl, pH 7.2). The elute was then centrifuged at 2600× *g* for 30 min (KUBOTA 2420 Compact tabletop centrifuge). Further, 2 mL of the pellet was resuspended in 5 ml of PBS at 40 °C. Viral DNA extraction was performed with the concentrated water sample using the MagPurix^®^ Viral Nucleic Acid Extraction Kit-V1.0 (ZP02003) and a fully automated MagPurix 12s nucleic acid extraction System (Zinexts Life Science Corp., New Taipei, Taiwan). The extracted final volume (100 μL) was analyzed for AdVs and NoVs by nested PCR.

The shellfish samples were randomly selected from individual sampling sites, and the oysters were dissected according to the previous protocols with some modifications [60,61]. Their digestive tissues were recovered from 10–12 shellfish samples and homogenized by ULTRA-TURRAX^®^ Tube Drive (IKA, Staufen, Germany). Then 2 ml of resulting homogenate was taken and added with 40 mL of PBS (7.5 mM Na_2_HPO_4_, 3.3 mM NaH_2_PO_4_, 108 mM NaCl, pH 7.2). This was shaken at 250 rpm for 2 hours at 4 °C. The treated homogenate was clarified by centrifugation at 2600× *g* for 30 min. Nearly 32.5 mL of the supernatant was taken and added with 3.3 mL NaCl (4M) and 8.1 mL PEG (50%). The resulting homogenate was shaken at 250 rpm for 16 h at 4 °C followed by centrifugation at 2600× *g* for 30 min. The pellet was discarded, and the supernatant was added with 250 μL PBS to re-dissolve the PEG precipitate. Then a 600 μL VB buffer (viral nucleic acid extraction kit) was added at room temperature and kept aside for 10 min. To precipitate the virus DNA, 750 μL absolute alcohol was added with the solution and centrifuged at 10,000× *g* for 1 min. The pellet was added with 400 μL of wash buffer1 (viral nucleic acid extraction kit) and centrifuged 10,000× *g* for 30 s, and this process was repeated twice. Then it was added with 50 μL of RNase-free water for 3 min at room temperature and again centrifuged at 10,000× *g* for a min. EasyPure Viral Nucleic Acid Extraction Kit (Bioman Scientific Co., LTD, Taipei, Taiwan) was used according to the operation manual to extract the viral nucleic acid. 

As NoVs is an RNA virus, the RNA must be reverse-transcribed into complementary DNA (cDNA). The extracted RNA was subjected to reverse transcription (RT) using a cDNA RT kit (Bosite Biotechnology Limited Company, Tianjin, China). The reaction mixture comprising 50 μL of RNA and 2.5 μL of random hexamers (1 μg/μL) was heated at 65 °C for 5 min to facilitate the opening of the RNA secondary structure. Next, the mixture was incubated with 20 μL of 5× reaction buffer, 2.5 μL of Moloney Murine Leukemia Virus (MMLV) reverse transcriptase (200 U/μL) (Protech Technology Enterprise Co., LTD, Taiwan), 2.5 μL of RNase inhibitor (40 U/μL), 10 μL of 10 mM dNTP pre-Mix and 12.5 μL of diethyl pyrocarbonate-treated water in ice. The PCR conditions (Px2 Thermal Cycler, Thermo, Waltham, MA, USA) were as follows: 25 °C for 10 min (random hexamer binding), 42 °C for 60 min (RT), and 70 °C for 10 min (termination of the reaction) [62]. cDNA stored at −20 °C until nested PCR analysis. 

### 4.3. Nested PCR, Product Sequencing, and Phylogenetic Analysis

For AdVs detection, the nested PCR mixture comprised 1 µL of each of the outer and inner primer sets (10 µM), 2 µL of DNA template, 5 µL of Fast-Run Tag Master Mix with dye (Protech Technology Enterprise Co., LTD., Taipei, Taiwan), and 17 µL of PCR-grade water. Hex1deg/Hex2deg and neHex3deg/neHex4deg are the outer and inner primer sets used to detect AdVs, respectively [63]. The PCR conditions were as follows: initial temperature of 95 °C for 5 min followed by 40 cycles of 95 °C for 30 s, 56 °C for 30 s, 72 °C for 30 s, and final of 72 °C for 5 min (Thermo Px2 Thermal Cycler, Waltham, MA, USA).

For NoVs detection, the nested PCR mixture comprises 3 µL of cDNA template, 1 µL of each of the outer and inner primers (10 µM), 5 µL of Fast-Run Tag Master Mix with dye (Protech Technology Enterprise Co., LTD., Taipei, Taiwan), and 16 µL of PCR-grade water. COG1F/G1-SKR and G1-SKF/G1-SKR were the outer and inner primer sets used to detect NoVs GI genotype, respectively [64,65]. COG2F/ G2-SKR and G2-SKF/ G2-SKR were the outer and inner primer sets used to detect the NoVs GII genotype, respectively [62,65]. The first PCR conditions were as follows: 95 °C for 4 min, followed by 40 cycles of 95 °C for 30 s, 56 °C for 30 s, and 72 °C for 1 min, a final extension step of 72 °C for 7 min, and hold at 4 °C. The second PCR conditions were as follows: 95 °C for 4 min, followed by 40 cycles of 95 °C for 30 s, 60 °C for 30 s, and 72 °C for 1 min, a final extension step of 72 °C for 7 min, and hold at 4 °C. 

The PCR products (5 µL) were mixed with 1 µL of DNA-loading dye and subjected to agarose gel electrophoresis using a 1.5% gel in Tris-acetate buffer (TAE) at 100 V for 30 min (Apelex P.S 304, USA). The resolved bands were visualized under a UV transilluminator, and the images were captured using a gel documentation system (Sankyo Denki G14T8-AN, Taiwan Rishun). The nested PCR products of AdVs and NoVs from the positive samples were sequenced. The PCR products were excised from the gel and purified. Further, the purified nested PCR products were sequenced using a 3730xl DNA analyzer (Applied Biosystems, Mission Biotech Co. Ltd., Taipei, Taiwan). The nucleotide sequences were compared with the available information in the National Center for Biotechnology Information (NCBI) GenBank database and PubMed NCBI BLAST program. The phylogenetic distances were calculated using the Maximum Composite Likelihood model and were later analyzed by the Neighbor-joining (NJ) method using MEGA software version 7.0 (MEGA software, USA).

### 4.4. Real-Time qPCR for Quantification of AdVs 

Quantization of AdVs was performed using an ABI StepOneTM Real-Time qPCR Systems (Applied Biosystems, USA). The inner and outer primer sets (JTVX-F and JTVX-R) and the TaqMan probe (JTVX-P) were used for the quantification of HAdVs [66]. The limit of quantification (LOQ) of the TaqMan assay is 5 genomic equivalent copies (GEC) for HAdV40 and 8 GEC for HADVs 41 [66]. The qPCR mixture comprised 3 μL of DNA template, 17 μL of reaction buffer containing 0.8 μL of outer and inner primer (10 μM), 10 μL of Fast-Run Taq Master Mix with Dye (Protech Technology Enterprise Co., LTD, Taipei, Taiwan), and 4.6 μL deionized water. The PCR conditions were as follows: initial temperature of 95 °C for 5 min, followed by 40 cycles at 95 °C for 10 s, 55 °C for 30 s, and 72 °C for 20 s. 

### 4.5. Statistical Analysis

A Mann-Whitney U test was calculated to analyze the correlation between the AdVs positive/negative samples and water quality indicators. All statistical analyses were performed using the R software (version 3.6.0).

## 5. Conclusions

This study reports the prevalence, seasonal occurrence, quantification, and genotypes of AdVs/NoVs present in the water and shellfish samples taken from the major fishing ports and coastal oyster breeding farms in Taiwan. The results express the importance of screening shellfish for major enteric viruses since AdVs/NoVs have been suggested as indicators of enteric virus present in contaminated water and food. A higher detection rate was recorded in the low-temperature period than in high-temperature. The predominance of HAdVs serotype 4, PAdVs serotype 5, NoVs GI.2, GI.3, and GII.2 in the water and shellfish samples can pose a potential public health challenge. The oyster breeding farms and fishing ports may pollute via significant quantities of fecal contaminants through the runoff of livelihood sewage and livestock farm wastewater. Moreover, the confluence of polluted river water into the coastal environment may lead to the transport of fecal contaminants. Subsequently, the accumulated viruses get retained and concentrated into the shellfish via their feeding behavior. Thus, it is suggested to frame better estuarine environmental controls to prevent AdVs/NoVs transmission pathways by directly monitoring estuarine areas for sewage discharge into the seawater. The results of this surveillance study will be useful for future risk-based assessments of the consumption of AdVs/NoVs contaminated shellfish. This will be essential for developing improved guidelines for shellfish quality.

## Figures and Tables

**Figure 1 pathogens-11-00316-f001:**
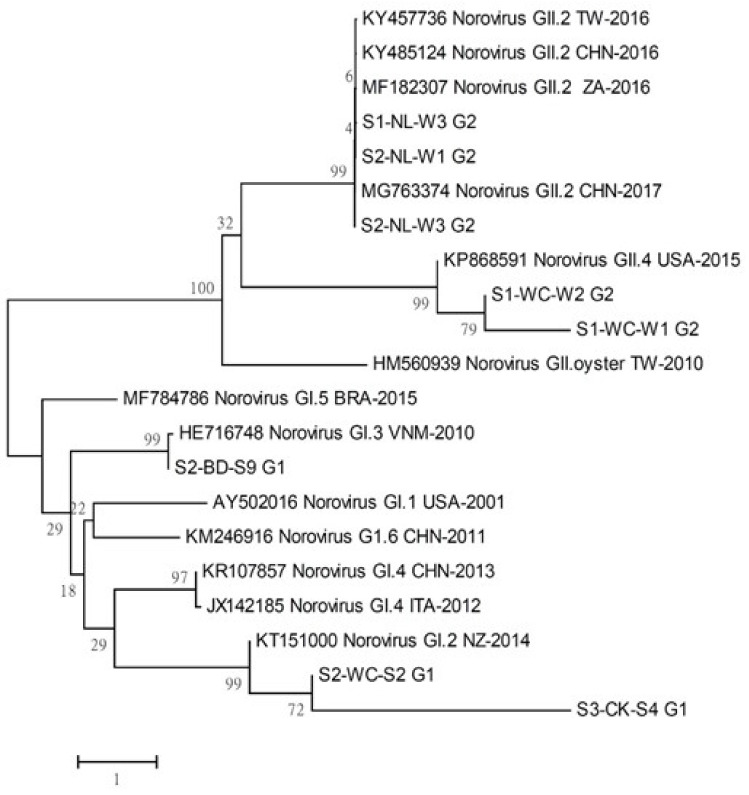
Phylogenetic analysis of NoVs in the water and shellfish samples across the fishing ports and coastal oyster breeding farms. Note: S indicates the period: S1 Low-temperature; S2 Warm-temperature; S3 High-temperature). Station name abbreviations: NL (Nanliao), WC (Wuchi), BD (Budai), CK (Chiku). W—water samples, S—shellfish samples).

**Figure 2 pathogens-11-00316-f002:**
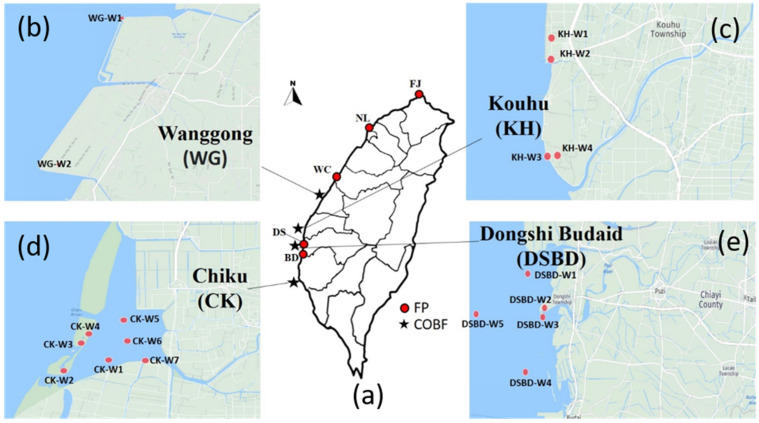
(**a**) Distribution of sampling sites across the major fishing ports (FP) and coastal oyster breeding farms (COBF) in Taiwan, (**b**) Distribution of sampling sites at Wanggong (WG) COBF, (**c**) Distribution of sampling sites at Kouhu (KH) COBF, (**d**) Distribution of sampling sites at Dongshi Budai COBF and (**e**) Distribution of sampling sites at Chiku (CK) COBF.

**Table 1 pathogens-11-00316-t001:** Detection and quantification of AdVs and NoVs in water samples across the major fishing ports and coastal oyster breeding farms.

Samples		LTP		WTP		HTP		Total		AdVs qPCR (Copies/L)		
		AdVs	NoVs	AdVs	NoVs	AdVs	NoVs	AdVs	NoVs	LTP	WTP	HTP
FP	FJ-W	33.3% (1/3)	0% (0/3)	0% (0/3)	0% (0/3)	0% (0/3)	0% (0/3)	11.1% (1/9)	0% (0/9)	-	-	-
	NL-W	0% (0/3)	33.3% (1/3)	100% (3/3)	66.6% (2/3)	0% (0/3)	0% (0/3)	33.3% (3/9)	33.3% (3/9)	-	5.25 × 10^4^	-
	WC-W	0% (0/3)	66.6% (2/3)	0% (0/3)	0% (0/3)	0% (0/3)	0% (0/3)	0% (0/9)	22.2% (2/9)	-	-	-
	DS-W	66.6% (2/3)	0% (0/3)	0% (0/3)	0% (0/3)	100% (3/3)	0% (0/3)	55.5% (5/9)	0% (0/9)	-	-	-
	BD-W	0% (0/3)	0% (0/3)	0% (0/3)	0% (0/3)	0% (0/3)	0% (0/3)	0% (0/9)	0% (0/9)	-	--	-
	Total	20% (3/15)	20% (3/15)	20% (3/15)	13% (2/15)	20% (3/15)	0% (0/15)	20% (9/45)	11.1% (5/45)	-	-	-
COBF	WG-W	50% (1/2)	0% (0/2)	50% (1/2)	0% (0/2)	100% (2/2)	0% (0/2)	66.6% (4/6)	0% (0/6)	6.03 × 10^3^	1.63 × 10^4^	-
	KH-W	100% (4/4)	0% (0/4)	25% (1/4)	0% (0/4)	50% (2/4)	0% (0/4)	58.3% (7/12)	0% (0/12)	7.94 × 10^4^	-	-
	DSBD-W	50% (5/10)	0% (0/10)	20% (2/10)	0% (0/10)	70% (7/10)	0% (0/10)	46.7% (14/30)	0% (0/30)	1.23 × 10^3^	1.53 × 10^4^	1.00 × 10^6^ 6.25 × 10^5^ 7.78 × 10^4^ 8.32 × 10^4^ 1.35 × 10^5^
	CK-W	42.8% (3/7)	0% (0/7)	0% (0/7)	0% (0/7)	0% (0/7)	0% (0/7)	14.3% (3/21)	0% (0/21)	-	-	-
	Total	56.5% (13/23)	0% (0/23)	17.4% (4/23)	0% (0/23)	47.8% (11/23)	0% (0/23)	40.6% (28/69)	0% (0/69)	-	-	-

W—Water sample; LTP—Low-temperature period; WTP—Warm-temperature period; HTP—High-temperature period; FP—Fishing port; COBF—Coastal oyster breeding farms.

**Table 2 pathogens-11-00316-t002:** Detection and quantification of AdVs and NoVs in shellfish samples across the major fishing ports and coastal oyster breeding farms.

Samples		LTP		WTP		HTP		Total		AdVs qPCR (Copies/100g)		
		AdVs	NoVs	AdVs	NoVs	AdVs	NoVs	AdVs	NoVs	LTP	WTP	HTP
FPM	FJ-S	0% (0/3)	0% (0/3)	33.3% (1/3)	0% (0/3)	0% (0/3)	0% (0/3)	11.1% (1/9)	0% (0/9)	-	4.27 × 10^4^	-
	NL-S	0% (0/3)	0% (0/3)	0% (0/3)	0% (0/3)	0% (0/3)	0% (0/3)	0% (0/9)	0% (0/9)	-	-	-
	WC-S	0% (0/3)	0% (0/3)	33.3% (1/3)	33.3% (1/3)	66.6% (2/3)	0% (0/3)	33.3% (3/9)	11.1% (1/9)	-	-	-
	DS-S	0% (0/3)	0% (0/3)	0% (0/3)	0% (0/3)	0% (0/3)	0% (0/3)	0% (0/9)	0% (0/9)	-	-	-
	BD-S	0% (0/9)	0% (0/9)	0% (0/9)	11.1% (1/9)	0% (0/9)	0% (0/9)	0% (0/27)	3.7% (1/27)	-	-	-
	Total	0% (0/21)	0% (0/21)	9.5% (2/21)	9.5% (2/21)	9.5% (2/21)	0% (0/21)	6.3% (4/63)	3.2% (2/63)	-	-	-
COBF	WG-S	0% (0/2)	0% (0/2)	0% (0/2)	0% (0/2)	0% (0/2)	0% (0/2)	0% (0/6)	0% (0/6)	-	-	-
	KH-S	0% (0/3)	0% (0/3)	0% (0/3)	0% (0/3)	0% (0/3)	0% (0/3)	0% (0/9)	0% (0/9)	-	-	-
	DSBD-S	30% (3/10)	0% (0/10)	20% (2/10)	0% (0/10)	0% (0/10)	0% (0/10)	16.7% (5/30)	0% (0/30)	-	7.03 × 10^3^	-
	CK-S	20% (1/5)	0% (0/5)	20% (1/5)	0% (0/5)	0% (0/5)	20% (1/5)	13.3% (2/15)	6.7% (1/15)	-	3.57 × 10^3^	-
	Total	20% (4/20)	0% (0/20)	15% (3/20)	0% (0/20)	0% (0/20)	5% (1/20)	11.7% (7/60)	1.7% (1/60)	-	-	-

S—Shellfish sample; LTP—Low-temperature period; WTP—Warm-temperature period; HTP—High-temperature period; FPM—Fishing port market; COBF—Coastal oyster breeding farms.

**Table 3 pathogens-11-00316-t003:** Distribution of various AdVs and NoVs genotypes in water and shellfish samples across the major fishing ports and coastal oyster breeding farms.

Samples		LTP		WTP		HTP	
		AdVs	NoVs	AdVs	NoVs	AdVs	NoVs
FP	FJ-W	H 41 (1/3)	-	-	-	-	-
	NL-W	-	GII.2 (1/3)	H 41 (3/3)	GII.2 (2/3)	-	-
	WC-W	-	GII.2 (2/3)	-	-	-	-
	DS-W	H 41 (1/3); P 5 (1/3)	-	-	-	P 5 (3/3)	-
	BD-W	-	-	-	-	-	-
COBF	WG-W	P 5 (1/2)	-	P 5 (1/2)	-	P 5 (2/2)	-
	KH-W	H 41 (1/4); P 5 (3/4)	-	P 5 (1/4)	-	P 5 (2/4)	-
	DSBD-W	H 41 (4/10); H 12 (1/10)	-	P 5 (2/10)	-	P 5 (7/10)	-
	CK-W	H 41 (2/7); H 12 (1/7)	-	-	-	-	-
FPM	FJ-S	-	-	H 41 (1/3)	-	-	-
	NL-S	-	-	-	-	-	-
	WC-S	-	-	H 41 (1/3)	GI.2 (1/3)	H 41 (2/3)	-
	DS-S	-	-	-	-	-	-
	BD-S	-	-	-	GI.3 (1/9)	-	-
COBF	WG-S	-	-	-	-	-	-
	KH-S	-	-	-	-	-	-
	DSBD-S	H 41 (2/10); H 12 (1/10)	-	H 41 (1/10); P 5 (1/10)	-	-	-
	CK-S	H 41 (1/5)	-	P 5 (1/5)	-	-	GI.2 (1/5)

W—Water sample; S—Shellfish sample; H—Human adenovirus; P—Porcine adenovirus; LTP—Low-temperature period; WTP—Warm-temperature period; HTP—High-temperature period; FP—Fishing port; COBF—Coastal oyster breeding farms; FPM—Fishing port market.

**Table 4 pathogens-11-00316-t004:** Nonparametric statistical analysis of the presence and absence of AdVs in relation to water quality parameters.

Water Quality Indicators	Mann-Whitney U Test	AdVs–Positive samples (*n* = 15)			AdVs–Negative Samples (*n* = 23)		
		Median	Q1	Q3	Median	Q1	Q3
Heterotropic plate count (CFU/ml)	*p* = 0.57	650.47	3.33	1753.33	1130.87	0.00	12306.66
Water temperature (°C)	*p* = 0.36	23.05	17.49	28.34	22.08	17.32	28.13
Turbidity (NTU)	*p* = 0.49	3.39	0.17	11.53	3.14	0.00	8.79
pH	*p* = 0.16	9.06	7.62	11.44	8.60	7.35	10.35
Salinity (%)	*p* = 0.64	20.18	0.99	31.78	20.92	0.01	32099.84
Dissolved oxygen (mg/L)	*p* = 0.44	14.39	0.38	39.66	10.88	0.42	33.58

AdVs for adenovirus; Q1 for first quartile; Q3 for third quartile.

## Data Availability

The data presented in this study are available on request from the corresponding author.

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
