# Peer review of "Surveillance of Adenovirus and Norovirus Contaminants in the Water and Shellfish of Major Oyster Breeding Farms and Fishing Ports in Taiwan"

_pathogens, 2022, doi:10.3390/pathogens11030316_

Round 1
Reviewer 1 Report
Line29: define abbreviation ‘fp’
Line 56: shellfish will retain and accumulate enteric viruses, please change the word ‘may’
Line 107: define abbreviations FP and COBF
Lines 120-127: define abbreviations
Define what is the low and what is the high temperature range
Line 189: Please add in the discussion a comment that it was unexpected that the most prevalent strains such as GII.4 and GII.17 were not detected, GI strains are linked to more foodborne outbreaks so logic that these were found.
Discussion: limitation of this study is that only 2 viruses were looked for, many other enteric viruses could have been checked.
Author Response
We are grateful to the reviewer for his insightful comments on our paper. We have been able to include changes to reflect most of the suggestions provided by the reviewer.

Reviewer 2 Report
The manuscript entitled “Surveillance of adenovirus and Norovirus contaminants in the water and shellfish of major oyster breeding farms and fishing ports in Taiwan” investigates the prevalence, seasonal occurrence, genetic diversity of Adenovirus and Norovirus along coastal farming, fishing ports and markets in Taiwan.
The manuscript is well written for the introduction and discussion sections but results and materials and methods need to be improved.
Here are some comments and mistakes that should be addressed.
Abstract:
pag 1 line 23: fishing ports, added the abbreviations FP because you used the abbreviation in line 29 without explaining it first.
Results
Pag 3 line 129: the term meagre is not scientific please change it with low.
Pag 4 line 189: Figure 2 is incorrect, please change with figure 1
Overall, the table of results is too complex and poorly understood, in my opinion it would be better to graph the data by separating by matrices (eg water, mussels) and by area (eg fishing port, COBF) for better readability. Furthermore, in the table as it is there is an error in the unit of measurement AdVs qPCR is in copies / L on page 4 but this applies to water, on page 5 the molluscs matrix has Copies / 100gr as unit of measurement but not it is indicated in the header because it is the same for all matrices for how the table is made. It is also unclear whether the quantitative value is a single sample or the mean. If on three samples AdV was found on all three the quantitative value to whom does it refer? just to a champion? therefore either the value is set for each one or the positives found are averaged; please clarify......
Additionally, sequencing results can be reported separately. There is too much information in the table all at once and it is illegible.
Discussion
References num 23 appears in materials and methods instead it must be here because the discussion comes first in the text of materials and methods, consequently all the numbers in the following must be changed, please correct them.
Pag 7 line 222: the term meagre is not scientific please change it with low.
Pag 8 line289: there is a parenthesis by itself.
Pag 9 line 303: there is a parenthesis by itself.
Pag 9 Line 312: there is a reference in words (Tao et al., 2016), it must be put in number and added in bibliography.
Pag 9 Line 316: lined is not correct, please change with linked
Pag 9 Line 352: there is a reference in words (Hundesa et al., 2006), it must be put the number 52 that is already present in bibliography.
Pag 9 Line 354: there is a reference in words (Bosch et al., 1998), it must be put in number and added in bibliography.
Materials and methods
Pag 10
line 359 why you used only 1 L for testing seawater? Usually for bathing water virus research is done with 10 L while 1L is tested for wastewaters samples (Wyn Jones et al., 2011. Surveillance of Adenovirus and noroviruses in European recreational waters)
line 364: “oyster breeding farms the” is repeated 2 times can be deleted.
Line 367: Fig 1 is not norrect change with Fig 2
Line 372: NL S-/21 is correct? the number of samples is missing
Line 375: traansferred is not correct, please change with transferred
Line 376: temeprature is not correct, please change with temperature
Figure 2 in maps (b). Wanggong is correct? In the text you wrote Wang Gong. In caption (line 378) remove of adenovirus and norovirus… this part is not needed in the sentence.
Line 387: reference 23 is not correct, this number must be put in Discussion, change the number based on progression.
Pag 11:
Section 4.2: Please added references regarding the methods used for virus concentration and pre- treatment of shellfish
Usually, water filtration with 45 microliter membranes is used for bacteria, while for viruses, membranes with lower porosity are used (e.g. ultrafiltration). Do you have a reference to add where this technique is shown to be correct for viruses as well? having lost many viral particles using membranes not suitable for the purpose.
Line 401- 402: 1.5 grams of hepatopancreas comes from a hepatopancreas pool of how many individuals? add this information.
Section 4.3: enter the concentrations of the reagents and primers used for the mix not only the quantity.
Pag 12 line 454: please add the kit used.
Line 455: what is the method of sequencing used? Please add some informations
Pag 12 section 4.4: What’is the limit of quantification (LOQ) of Real Time qPCR method for AdVs used?
Conclusions
Pag 12 Line 477: Add in Taiwan after breeding farms.
References
Add Tao et al., 2016; Bosch et al., 1998 and Jothikumar et al., 2005.
Author Response

(The authors gave the same response as above.)

Round 2
Reviewer 2 Report
The revised article “Surveillance of adenovirus and Norovirus contaminants in the water and shellfish of major oyster breeding farms and fishing ports in Taiwan” investigates the prevalence, seasonal occurrence, genetic diversity of Adenovirus and Norovirus along coastal farming, fishing ports and markets in Taiwan” is more understandable and clearer than the previous version. However, there are still some issues in the manuscript that need to be modified.
Results
Pag 3 line 111: after table 1 add… and table 2.
Pag 4 line 192: Fig. according to author’s guidelines change with Figure.
Pag 5 in table 1 add a horizontal line after the first total to separate FP from COBF so it's more readable
Pag 6 in table 2 add a horizontal line after the first total to separate FPM from COBF so it's more readable
Materials and methods
Pag 11 line 398: Fig. according to author’s guidelines change with Figure.
Pag 12 line 424: there is a parenthesis by itself.
Author Response
I thank you for your thoughtful suggestions and insights. The manuscript has benefited from these insightful suggestions.
